# Effects of Different Drying Methods on Drying Kinetics, Microstructure, Color, and the Rehydration Ratio of Minced Meat

**DOI:** 10.3390/foods8060216

**Published:** 2019-06-18

**Authors:** Aslı Aksoy, Salih Karasu, Alican Akcicek, Selma Kayacan

**Affiliations:** Department of Food Engineering, Faculty of Chemical and Metallurgical, Yildiz Technical University, Esenler, İstanbul 34210, Turkey; aksoyas@gmail.com (A.A.); akcicekalican82@gmail.com (A.A.); selmakayacan@gmail.com (S.K.)

**Keywords:** ultrasound assisted drying, minced meat, SEM, microstructural

## Abstract

This study aimed to investigate the effect of different drying methods, namely ultrasound-assisted vacuum drying (USV), vacuum drying (VD), and freeze-drying (FD), on the drying kinetics and some quality parameters of dried minced meat. In this study, USV was for the first time applied to the drying of minced meat. The USV and VD methods were conducted at 25 °C, 35 °C, and 45 °C. The different drying methods and temperatures significantly affected the drying time (*p* < 0.05). The USV method showed lower drying times at all temperatures. The rehydration values of the freeze-dried minced meat samples were higher than those obtained by the USV and VD techniques. The samples prepared using USV showed higher rehydration values than the vacuum dried samples for all temperatures. The effects of the different drying techniques and drying conditions on the microstructural properties of the minced meat samples were investigated using scanning electron microscope (SEM). The USV method resulted in higher porosity and a more open structure than the VD method. Total color differences (Δ*E*) for VD, USV, and FD were 8.27–20.81, 9.58–16.42, and 9.38, respectively, and were significantly affected by the drying methods and temperatures (*p* < 0.05). Higher drying temperature increased the Δ*E* value. Peroxide values (PV) significantly increased after the drying process, and samples treated with USV showed lower PV values than the VD treated samples. This study suggests that USV could be used as an alternative drying method for minced meat drying due to lower drying times and higher quality parameters.

## 1. Introduction

Meat and meat products are valuable and perceived as high-quality foods because they contain dietary protein; vitamin B; minerals, especially zinc, iron, and phosphorus; and essential fatty acids in high amounts [1,2]. However, meat and meat products are susceptible to microbial growth because of their high water activity and nutrient composition [3]. Therefore, meat products, such as minced meat, are recognized as a highly perishable food with a limited shelf-life during processing, transportation, and storage. Minced meat should be subjected to preservation methods to increase its shelf-life.

The drying process is one of the oldest and most widely used preservation methods because it is efficient and cheap. Drying extends the shelf-life of meat and meat products because of the reduction in water activity [4]. In addition, the drying process reduces the microbial load and results in lighter volumes and weights, which decrease storage and transportation costs [5]. Dried meat that has been sliced or cut into cube-shaped pieces is used in the food industry as an ingredient in different formulations, such as paste products and instant soup formulations, to improve the nutritional and sensory value of products [6].

In conventional thermal drying methods, flavor, color, and nutritional loss (vitamin degradation and loss of amino acids) occur because of thermal degradation, which decreases the drying rate and rehydration ratio [5]. For this reason, new technologies should be used to obtain better quality dried meat and dried meat products. Drying methods and processing conditions affect the properties of dried meat products, such as the porous structure, shrinkage, and bulk density [7]. In this study, the modern drying techniques of vacuum drying (VD), ultrasound-assisted vacuum drying (USV), and freeze-drying (FD) were used to dry minced meats. These methods show some advantages during the drying process. In VD, the use of low temperatures in the absence of oxygen can preserve heat-sensitive and easily oxidizable foods. Thus, discoloration and the decomposition of the flavor and some nutritional substances can be prevented [8,9]. However, the quality loss of dried foods is not fully prevented using VD. Therefore, some other techniques, such as ultrasound, microwave, and radio frequency, might be applied to reduce drying time and quality loss.

To shorten the drying time, ultrasonic treatment and vacuum drying are combined in the USV technique [10]. Using this technique, the vacuum process, by disrupting the cell walls of the food, accelerates water transfer, which increases the drying rate. Mechanical waves produced by the ultrasonic treatment facilitate the transfer of heat and water from inside to the surface of the food. Transfer rates change according to the pressure and frequency of the sonic waves [10]. Cavitation, which facilitates water transfer, is another important mechanism applied during ultrasound drying because it results in the removal of firmly attached water [11]. The ultrasonic enhancement of food drying can be summarized as an increased drying rate, the prevention of damage to food quality, a reduction in energy consumption and process cost, and the modification of the composition of the food [11].

FD is one of the best food drying methods for biological materials sensitive to heat and oxidation. It does not cause protein denaturation or loss of vitamins, and therefore freeze-dried products retain their nutritional value [12]. Using FD, products have a porous structure, and good rehydration capability, taste, and flavor are possible [13].

Besides selecting the correct drying method, optimization of the drying temperature is another important parameter affecting the quality of dried food products. High temperatures provide low drying times due to more rapid heat and moisture transfer. However, high temperatures lead to the degradation of some heat sensitive components, such as protein and vitamins, as well as reduce color quality and rehydration ability, and increase shrinkage rates [7,14,15]. Low-temperature drying processes are suggested for drying meat products because they reduce microbiological and biochemical decomposition [16]; however, at very low temperatures, the drying time may be prolonged. A drying temperature that does not reduce the physicochemical, nutritional, and sensory quality of a product and provides a reasonable drying time should be selected.

The main aim of this study was to determine the effects of different drying techniques on the quality of minced meat. For this purpose, the drying of minced meats was conducted using different drying methods at different temperatures and the effects on the quality of the minced meat were evaluated. In this study, VD (at 25 °C, 35 °C, and 45 °C), with and without ultrasound pretreatment, and FD methods were used to determine the physicochemical properties of the minced meat.

## 2. Materials and Methods

### 2.1. Materials 

Minced meat samples (from beef meat) were purchased from a local marketplace in Istanbul, Turkey and brought to the laboratory at 4 °C. The samples were stored in the refrigerator until analysis and drying process. Minced meat was spread on a silicon plate with a diameter of 9 cm and a thickness of 5 mm (21 ± 0.5 g) before the drying process.

### 2.2. Methods

The research was carried out in two steps. First, the minced meat was dried by different drying techniques, and then the rehydration ratio, shrinkage, color measurement, and microstructure of dried minced meat were determined. All measurements were carried out in triplicate.

#### 2.2.1. Physicochemical Properties of Minced Meats

The dry matter content of meat samples was determined by the procedure of AOAC 950.46 [17]. The drying process was performed using an oven drier (Memmert UF110, Munich, Germany) at 105 °C to achieve constant weight. The drying process was finished when a constant weight of dried minced meat was reached. The fat content of the minced meat samples was determined according to the AOAC 960.39 [18]. A Soxhlet extraction system (Daihan, WHM-12293, Gangwon-do, South Korea) was used for fat extraction. Hexane was used as a solvent for fat extraction. The peroxide values (PV) of the oils were determined according to the methodology of the IUPAC 2.50. The PV was expressed as mill equivalents of peroxide oxygen per kg of sample (mEq/kg). 

#### 2.2.2. Drying of Minced Meats with Different Techniques

The different drying techniques, namely ultrasonic vacuum (USV), vacuum (VD), and freeze dryer (FD), were used to perform the drying process of minced meat. The drying temperatures of 25 °C, 35 °C, and 45 °C were selected for USV and VD. 

The vacuum drying technique was carried out according to the method described by Tekin et al. [10] using a laboratory scale drying unit (Daihan WOV-30, Gangwon-do, South Korea). For vacuum drying, the vacuum was regulated by a vacuum pump (EVP 2XZ-2C, Zhejiang, China) with 60 mbar ultimate pressure and 2 L/s pump speed. The USV technique was performed according to the method described by Baslar, et al. [19]. An ultrasonic vacuum water bath (Daihan, WUC-D10H, Gangwon-do, South Korea) was used for the USV. The freeze drying technique was performed using a freeze dryer (Martin Christ, Beta 1-8 LSC plus, Osterode am Harz, Germany) at −55 °C and 1 hPa for 72 h. For all drying methods, the samples (about 21 g) were weighed using an analytical balance (Shimadzu, TW-423, Kyoto, Japan) with an accuracy of 0.1 mg. at every 15–30 min during the drying process until water content decreased to 12 ± 0.1% (*w/w*). 

#### 2.2.3. Drying Characteristic Analyses

##### Moisture Ratio

The moisture ratio (MR) of minced meat samples was calculated according to Equation (1):(1)$$MR=\frac{M_{t}-M_{e}}{M_{0}-M_{e}},$$where, MR is the moisture ratio, *M*_t_, *M*_0_ and *M*_e_ are, respectively, moisture content at each measurement time, initial moisture content, and equilibrium moisture content (kg water/kg dry matter). The relative moisture of the drying air (Equation (1)), which varied continuously during the drying experiments, is simplified into Equation (2) [20]:(2)$$\frac{M_{t}-M_{e}}{M_{0}-M_{e}}\;to\;\frac{M_{t}}{M_{0}},$$

##### Effective Moisture Diffusivity (*D*_eff_) and Activation Energy 

Fick’s second law of diffusion, Equation (3), is widely used to describe the effective spread of moisture (*D*_eff_) during the drying of most food materials. Equation (3):(3)$$K=\frac{\pi^{2}D_{\mathrm{eff}}}{4L^{2}}\cdot \frac{\partial M}{\partial t}=D_{\mathrm{eff}}\nabla^{2}M,$$where, *D*_eff_ is the effective moisture diffusivity (m^2^/s), *L* is the half-thickness of the slab in samples (m), and t is the drying time (s). Equation (3) uses an analytical solution and neglecting shrinkage, constant temperature, and diffusion contribution, as well as uniform starting moisture distribution. Equation (4) [21]:(4)$$MR=\frac{8}{\pi^{2}}\sum_{n=0}^{\infty }\frac{1}{{\left(2n+1\right)}^{2}}\exp\left(-{\left(2n+1\right)}^{2}\frac{\pi^{2}D_{\mathrm{eff}}t}{4L^{2}}\right),$$where, MR is the moisture ratio, *t* drying time, *D*_eff_ effective diffusivity (m^2^/s), *n* is the number of terms of the Fourier series, and *L* slices are half slab thickness (m). For long drying times Equation (4) can be further simplified in Equation (5) [22]:(5)$$MR=\frac{8}{\pi^{2}}\exp\left[\frac{-\pi^{2}D_{\mathrm{eff}}t}{4L^{2}}\right],$$

The *D*_eff_ values can be calculated by plotting the ln(*MR*) value as a function of time, and a straight line with a slope of (*K*) can be calculated from this plot, which can be written as Equation (6):(6)$$K=\frac{\pi^{2}D_{\mathrm{eff}}}{4L^{2}},$$

The Arrhenius equation is generally used to determine the temperature dependency of the *D*_eff_ values Equation (7): (7)$$D_{\mathrm{eff}}=D^{0}{}_{\mathrm{eff}}\exp\left(\frac{-E_{a}}{RT}\right),$$
where, *D*_0_ represents the pre-exponential factor of the Arrhenius equation (m^2^/s), *E*_a_ is the activation energy (kJ/mol), *R* is the ideal gas constant (kJ/mol·K), and *T* is the drying temperature in Kelvin [22]. 

#### 2.2.4. Rehydration Ratio of Dried Minced Meats

The rehydration of the minced meat was performed according to a modified method described by [23]. Two grams of dried minced meat samples were rehydrated in 20 mL distilled water at 30 °C for 20 min until a constant water content was reached. The rehydration ratio is described as the ratio of sample weight after and before rehydration. The rehydration ratio was then calculated by Equation (8) described below [23]: (8)$$RR=\frac{M}{M_{0}},$$
M: sample weight after rehydration, $M_{0}$: sample weight before rehydration.

#### 2.2.5. Shrinkage Ratio of Dried Minced Meats

Surface change% (shrinkage) is a common widely-known phenomenon during drying. Dimensional shrinkage of the dried minced meat samples was determined using the following Equation (9). The diameter and thickness of the fresh and dried minced meat samples were measured by using a steel Vernier caliper.
(9)$$Shrinkage\;\left(\%\right)=\left(\frac{\left(Raw\;thickness-dried\;thickness\right)+\left(Raw\;diameter\;-\;dried\;diameter\right)}{\;\left(Raw\;thickness\;+\;Raw\;diameter\right)}\right)\times 100,$$

#### 2.2.6. Color Measurement

##### Surface Color

The effect of different drying methods and temperatures on color changes of dried minced meats were determined at room temperature using a chroma meter (CR-13, KONICA MINOLTA, Tokyo, Japan) at four different edge spots on the surface of each sample before and after the drying treatment. The color evaluation procedure was based on the determination of Hunter values *L** (whiteness/darkness), *a** (redness/greenness), and *b** (yellowness/blueness).

To describe the change in the color values of samples, the total color difference (Δ*E*) value was calculated according to the following Equation (10):(10)$$\Delta E=\sqrt{{\left(\Delta L\right)}^{2}+{\left(\Delta a\right)}^{2}+{\left(\Delta b\right)}^{2}}$$

The higher the Δ*E* value, the greater the difference between the two measured samples.

#### 2.2.7. Microstructure

The images of dried minced meat samples were captured using a scanning electron microscope. The microscopic structures of the samples before and after rehydration were examined, and the morphologies and structures of the samples were observed by a field emission scanning electron microscope (SEM) (EVO LS 50, Zeiss, Feldbach, Switzerland). Dried minced meat samples were cut and fixed on the SEM stub and the samples were covered with a golden coat to provide a reflective surface for the electron beam. The gold-plated samples were then imaged under a microscope at 5 kV [19].

#### 2.2.8. Statistical Analysis

The measurements were repeated twice with three replications. Collected data were subjected to analysis of variance (ANOVA) using JMP 9 (SAS, Cary, NC, USA). When a significant (*p* < 0.05) main effect was found, mean values were further analyzed using the Duncan’s Multiple Range Test comparison test. 

## 3. Results and Discussion

### 3.1. Drying Characteristics of Minced Meats

Figure 1 presents the drying curves (time versus MR) obtained using ultrasonic vacuum and VD at 25 °C, 35 °C, and 45 °C. As shown in Figure 1, the MR showed a rapid decline over time in the initial drying period, which indicated that during this drying period, both USV and VD achieved a constant drying rate. This result was consistent with previously published studies [19,24,25]. In addition, with elapsed time, the drying rate decreased, which indicated that both drying methods showed a falling drying period rate.

The total drying time differed significantly according to both the drying method and the temperature. The shortest drying time was observed for the USV technique at 45 °C (82 min); this time was approximately 20% shorter than the time observed for VD at 45 °C (100 min). The constant rate periods for the USV and VD were 60 min and 70 min, respectively. The total drying times at 25 °C for the USV and VD were 165 min and 195 min, respectively, and the drying times at 35 °C were 115 min and 125 min for the USV and VD, respectively. Resulting from the ultrasound treatment, the diffusion boundary layer was changed, and this change is explained by the properties of the acoustic waves caused by the ultrasound [14,26]. 

Baslar, et al. [14] used an ultrasonic vacuum drying method to reduce the drying time of chicken and beef. In the current study, a shorter drying period was observed using USV, and ultrasound accelerated the drying process. Therefore, our findings are compatible with the literature [10,19,24].

As shown in Figure 1, the increase in the drying temperatures significantly decreased the drying times. The higher drying rates obtained at higher temperatures for both the USV and VD could have been due to the increase in water vapor pressure as a result of the increased heat and mass transfer rate. The forces and effects generated by ultrasonic waves in solid materials create an effect similar to that of a sponge. As a result of this sponge effect, liquid is released from the interior of particles to the solid surface. The forces involved in this system result from the ultrasonic waves and are generally greater than the surface tension forces [14]. All these factors affect the internal resistance to mass transfer. Reducing drying times is very important for the food industry. Fatty oxidation occurs as a result of the heat treatment of fat-containing food products, especially meat, and results in serious losses in the quality and shelf-life of meat and meat products. Therefore, the negative effect of fat oxidation can be reduced by using an ultrasonic vacuum drying technique. High drying temperatures result in high drying rates, which lead to a rapid reduction in moisture content and reduced drying time; some studies in the literature support this study [14,27].

### 3.2. Effect of Temperature and Different Drying Techniques on Effective Moisture Diffusivity (D_eff_) 

Effective moisture diffusivity (*D*_eff_), an important concept regarding physical and thermal properties, is defined as the transport of moisture at distant rates during the drying of food. As seen in Figure 2 and Figure 3, the *D*_eff_ values increased with increasing temperature. The *D*_eff_ values were calculated as 1.07 × 10^−8^–2.96 × 10^−8^ m^2^/s for the USV and 1.10 × 10^−8^–2.72 × 10^−8^ m^2^/s for the VD. These results were in agreement with previously published studies conducted on different food materials [19,28]. Comparisons of the *D*_eff_ values of the USV and vacuum drying techniques showed that the *D*_eff_ value for the USV was larger than the vacuum drying technique *D*_eff_ value at 35 °C and 45 °C (*p* < 0.05).

### 3.3. Peroxide Formation 

Table 1 shows the peroxide values of the fresh and dried samples of the minced meat. As shown, PV formation was significantly affected by the drying temperature and drying method, and the highest level (10.42) of peroxide formation was observed in the freeze-dried samples. The PV values for the VD and USV samples were 5.71–6.91 and 5.02–5.63, respectively. Similar results were reported by [29] for meat and tuna meat [30]. In these studies, greater peroxide formation was observed in freeze-dried meat products as compared with hot-air dried products. Rahman, Al-Amri and Al-Bulushi [30] reported that higher porosity and surface area led to an increase in the diffusion of oxygen from the surface to inner parts of the meat products, and drying temperature significantly increased the PV value. This could be explained by the increased thermal oxidation rate of polyunsaturated fatty acids with increasing temperature. The activation energy of the oxidation might have been reduced, leading to the breakdown of hydroperoxide into free radicals [31].

### 3.4. Rehydration and Microstructural Properties and Shrinkage Values

Table 2 shows the rehydration ratios of the dried samples. As shown, the drying temperature and applied method significantly affected the rehydration ratio (*p* < 0.05). The rehydration ratios of the freeze-dried minced meat samples were higher than those obtained by the VD and USV techniques. The USV samples showed higher rehydration values than the VD samples. This is explained by the formation of a porous structure resulting from the ultrasonic process. The highest rehydration values for the VD and USV techniques were obtained at 35 °C. This result is explained by the food matrix having less deformation due to moderate drying temperature and drying time. Similar to our results, in a study [20] on the effect of different drying techniques on some quality parameters of mushrooms, the best rehydration ratio was obtained from FD. According to their results, the rehydration ratio of ultrasound drying was higher than conventional methods. The authors concluded that the application of ultrasound power (using a 20 kHz probe) could lead to the development of greater internal stresses and the creation of pores. A study on the effect of ultrasound pretreatment on some quality parameters of carrot slices reported that larger rehydration ratios were observed for samples subjected to ultrasonic pretreatment [32]. The positive effect of ultrasound on different food materials has also been reported by other published studies [33,34]. As a result of the information obtained, the rehydration characteristics of the minced meat samples were preferred by the FD and alternatively by the low-temperature USV. The higher the rehydration rate the greater the water absorption, which resulted in a product closer to the original sample.

The shrinkage properties of the minced meats are presented in Table 2. As shown in Table 2, the shrinkage ratio increased with temperature for the vacuum and USV techniques. The increase in shrinkage values at higher temperatures was also reported by [35,36]. The vacuum-dried meat showed higher shrinkage values than the USV technique dried samples at the same temperature. This is explained by the ultrasonic treatment reducing the drying time and less shrinkage occurring in the samples dried using the USV technique. Liu, et al. [37] reported a lower shrinkage value after ultrasound pretreatment for the drying of eucalyptus.

Low shrinkage values are very important for dried minced meat samples. As shown in Table 1, the shrinkage ratios of the freeze-dried minced meat samples had lower values than those obtained by the VD and USV techniques. The lower the shrinkage rate, the more stable the microstructural properties of dried minced meat. The glass transition theory, among others, explains shrinkage during drying-related processes. According to this theory, greater numbers of pores, or less collapse, is observed in a material if it is subjected to temperatures below the glass transition temperature. In FD, where the drying temperature is below the glass transition temperature, the resulting shrinkage is very low, and the dried product has a highly porous structure. In hot air drying, the drying temperature is above the glass transition temperature and substantial shrinkage occurs in dried samples [38].

The effect of different drying techniques and conditions on the microstructural properties of minced meat samples was observed using SEM (Figure 4). As shown in Figure 4, different drying techniques were applied at different temperatures, and different outcomes were observed in the results. The dried samples obtained using the USV technique had a more open structure and greater porosity than the samples dried using the VD technique. For the USV technique, less shrinkage was observed in the samples as compared with the VD technique due to the shorter drying time and the evaporation of water on the inner surface. However, with an increase in temperature for both drying techniques, shrinkage occurred because of the closure of the surface of the porous structures due to oil. This situation is not desirable in dried meat products. In addition, as seen in Figure 4, the porosity and open structure of the freeze-dried minced meat samples were better than with the VD and USV techniques. The main reason for this is that the porosity and open structure were less damaged at low temperatures. Similar results have also been reported by other studies [39]. Rajewska and Mierzwa [40] reported that an ultrasound treatment led to microchannels in, and the disruption of, onion tissue, which increased the porosity and intercellular spaces.

### 3.5. Color Values of Dried Minced Meats

The Hunter color scale was used to evaluate the color characteristics of the minced meat samples dried by using different drying techniques. As shown in Table 3, the dried minced meat sample with the vacuum drying technique at 45 °C exhibited the highest Δ*E* value, which indicates the color difference between the fresh and dried samples. Moreover, this means that high temperatures over a long time period cause loss of color and a darker appearance. As shown in Table 2, USV at 45 °C causes less change in color properties as compared with VD and the color properties of the minced meat samples are better retained with ultrasonic vacuum drying. The color changes during drying are caused by the oxidation, change in surface structure of meat, and non-enzymatic browning reactions. As shown in Figure 4 and Figure 5, color loss occurs with the increase in temperature using vacuum drying and USV drying. In the FD dried samples, the color was better maintained as compared with VD and USV drying techniques. The Hunter *L** values of the FD dried samples were higher than the samples dried with other methods. A possible explanation is that high pore structures in the FD treated samples led to more uniform light reflection from the surface, and therefore a relatively translucent object was obtained [29]. Therefore, the color can be retained better for minced meat samples by the freeze-drying technique. Amami, Khezami, Mezrigui, Badwaik, Bejar, Perez and Kechaou [33] reported that lower color change was observed in the ultrasound-assisted drying as compared with conventional drying methods. Lower color change by ultrasound assisted process was also reported by Mendez-Calderon, et al. [41]. In our study, the lower color change in USV as compared with VD could be due to the low drying time at all temperatures. 

## 4. Conclusions

In this study, the effects of different drying temperatures and drying methods on the drying characteristics and quality parameters of minced meat samples were investigated and a USV technique was used for drying minced meat for the first time. The results of the study indicate that the drying methods and parameters significantly affected the drying time and quality characteristics. Drying time significantly decreased with increased drying temperature. The FD technique resulted in higher rehydration ratios and lower shrinkage, oxidation (PV) rates, and color change. The USV technique resulted in lower drying times, less color change and shrinkage, and a high porosity structure as compared with conventional VD. This study suggests that the USV technique is a useful method for reducing the drying time and increasing the quality parameters of minced meat.

## Figures and Tables

**Figure 1 foods-08-00216-f001:**
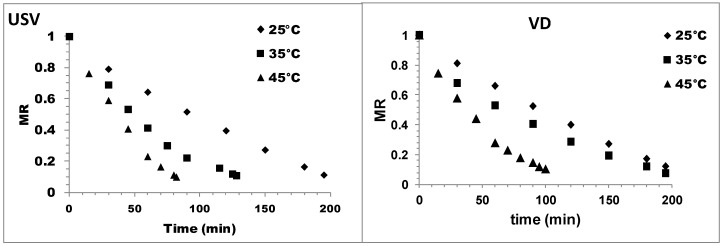
Drying characteristic of minced meat at different temperatures for ultrasound vacuum drying (USV) and vacuum (VD).

**Figure 2 foods-08-00216-f002:**
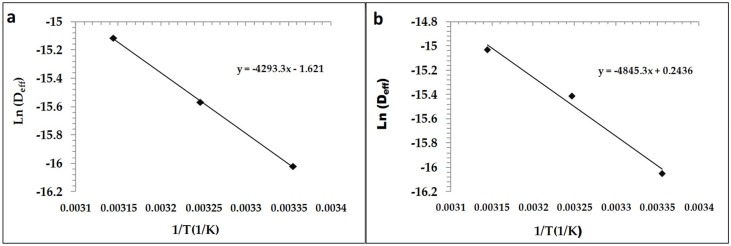
Arrhenius type relation between *D*_eff_ and reciprocal temperature (**a**: VD, **b**: USV).

**Figure 3 foods-08-00216-f003:**
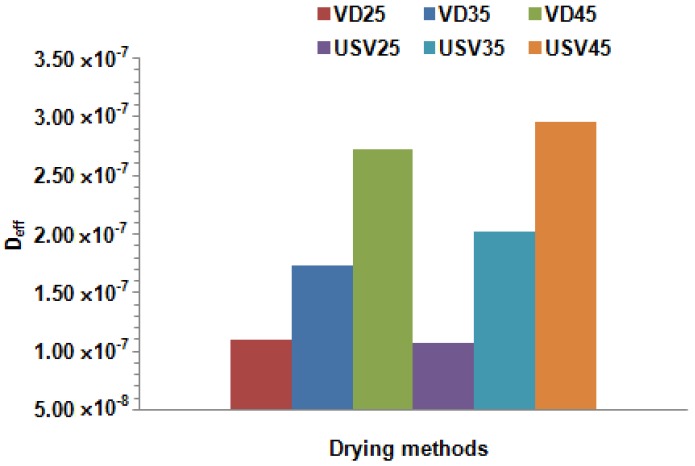
Effect of drying methods and temperatures on *D*_eff_ value.

**Figure 4 foods-08-00216-f004:**
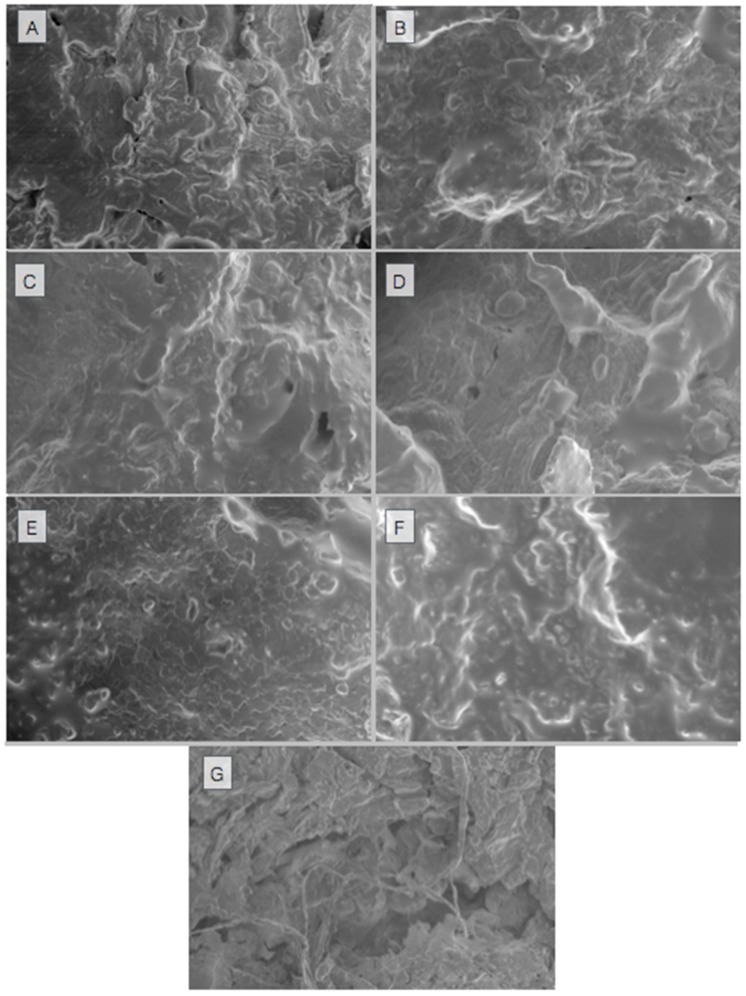
SEM images of the minced meat samples dried at different temperatures and drying methods (**A**, 25 V; **B**, 25 USV; **C**, 35 V; **D**, 35 USV; **E**, 45 V; **F**, 45 USV; **G**, FD).

**Figure 5 foods-08-00216-f005:**
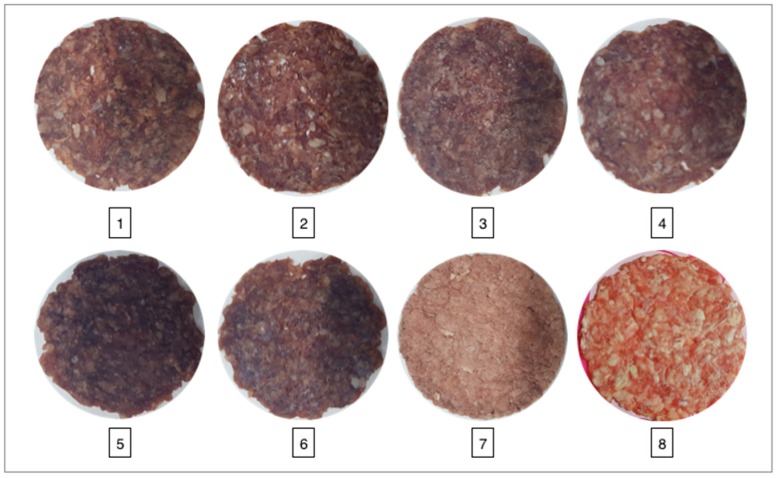
Pictures of the minced samples (**1**, 25 V; **2**, 25 USV; **3**, 35 V; **4**, 35 USV; **5**, 45 V; **6**, 45 USV; **7**, FD; **8**, fresh).

**Table 1 foods-08-00216-t001:** Peroxide value (PV) of the fresh and dried samples.

Drying Method	PV (mEqO_2_/kg)		
	25 °C	35 °C	45 °C
Fresh	0.96 ± 0.02 ^D^	0.96 ± 0.02 ^D^	0.96 ± 0.02 ^D^
VD	5.71 ± 0.04 ^cB^	5.97 ± 0.04 ^bB^	6.91 ± 0.06 ^aB^
USV	5.02 ± 0.01 ^bC^	5.61 ± 0.01 ^aC^	5.63 ± 0.02 ^aC^
FD	10.42 ± 0.29 ^A^	10.42 ± 0.29 ^A^	10.42 ± 0.29 ^A^

Different lowercase letters in the same line show statistical differences between drying methods (*p* < 0.05). Different uppercase letters in the same column show statistical differences between drying parameters (*p* < 0.05). VD, vacuum drying; USV, ultrasound assisted vacuum drying; FD, freeze drying; PV, peroxide value.

**Table 2 foods-08-00216-t002:** Effect of the freeze, vacuum, and ultrasound vacuum drying on shrinkage and rehydration value of minced meat.

Methods	Shrinkage			RH *		
	25 °C	35 °C	45 °C	25 °C	35 °C	45 °C
VD	22.14 ± 0.07 ^Ca^	30.00 ± 1.52 ^Ba^	37.14 ± 0.71 ^Aa^	1.40 ± 0.09 ^Bc^	1.45 ± 0.02 ^Ac^	1.27 ± 0.09 ^Cc^
USV	21.42 ± 0.06 ^Aa^	25.71 ± 0.58 ^Bb^	29.28 ± 0.87 ^Ab^	1.49 ± 0.04 ^Bb^	1.54 ± 0.05 ^Ab^	1.31 ± 0.03 ^Cb^
FD	17.14 ± 1.26 ^b^	17.14 ± 1.26 ^c^	17.14 ± 1.26 ^c^	1.93 ± 0.05 ^a^	1.93 ± 0.05 ^a^	1.93 ± 0.05 ^a^

Different lowercase letters in the same column show statistical differences between drying methods (*p* < 0.05). Different uppercase letters in the same line show statistical differences between drying parameters (*p* < 0.05). * RH: rehydration ratio.

**Table 3 foods-08-00216-t003:** Effect of different drying methods and temperature on the color quality of minced meat.

Drying Method	Δ*L*			Δ*a*			Δ*b*			Δ*E*		
	25 °C	35 °C	45 °C	25 °C	35 °C	45 °C	25 °C	35 °C	45 °C	25 °C	35 °C	45 °C
VD	6.04 ± 0.07 ^cB^	10.00 ± 1.41 ^bB^	16.18 ± 0.10 ^aA^	4.39 ± 0.09 ^bB^	2.69 ± 0.02 ^cB^	9.73 ± 0.15 ^aA^	3.55 ± 0.07 ^cA^	4.55 ± 0.09 ^bA^	8.75 ± 0.09 ^aA^	8.27 ± 0.06 ^cB^	11.31 ± 0.03 ^bA^	20.81 ± 0.05 ^aA^
USV	7.8 ± 0.03 ^bA^	11.98 ± 0.10 ^aA^	12.18 ± 1.44 ^aB^	4.04 ± 0.07 ^bB^	0.74 ± 0.00 ^cC^	7.91 ± 0.09 ^aB^	3.82 ± 0.01b ^A^	3.05 ± 0.09 ^bB^	7.67 ± 0.10b ^B^	9.58 ± 0.09 ^cA^	12.38 ± 1.04 ^bA^	16.42 ± 1.28 ^aB^
FD	6.64 ± 0.01 ^B^	6.64 ± 0.01 ^C^	6.64 ± 0.01 ^C^	6.09 ± 0.01 ^A^	6.09 ± 0.01 ^A^	6.09 ± 0.01 ^C^	2.48 ± 0.00 ^B^	2.48 ± 0.00 ^C^	2.48 ± 0.00 ^C^	9.34 ± 0.80 ^A^	9.34 ± 0.80 ^B^	9.34 ± 0.80 ^C^

Different uppercase letters in the same column show statistical differences between drying methods (*p* < 0.05). Different lowercase letters in the same line show statistical differences between drying parameters (*p* < 0.05). VD, vacuum drying; USV, ultrasound assisted vacuum drying; FD, freeze drying.

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
