# Peer review of "Effects of Different Drying Methods on Drying Kinetics, Microstructure, Color, and the Rehydration Ratio of Minced Meat"

_foods, 2019, doi:10.3390/foods8060216_

Reviewer 1 Report

Some words are misspelled, errors are most often in abstract. 

Line 14: affetcted, correctly affected

Line 15: temperutes, correctly temperatures

Line 141: question mark in denominator of equation formula

Line 152: unfinished sentence (...measured col...)

Line 289: draying, correctly drying

etc.

Author Response

Response to Reviewer 1 Comments

 Some words are misspelled, errors are most often in abstract. 

 Thank you very much for your suggestion for improving of our manuscript. The manuscript was extensively revised. All figures were reorganised. The figure caption and abbreviation was revised. The Table 2 (Table 3 in revised version) improved by providing standart deviation and statistic. All abbreviation was checked and corrected (for example Ultrasound assisted vacuum dryin is USV, Vacumm drying VD and freeze drying FD). Color figures reorganised and quality of the figures was improved. Peroxide analysis results were presented and discussed. Introduction part and discussion part were improved. Manuscript was sent to language editing.

 Point 1: 
Line 14: affetcted, correctly affected.

 Response 1: The revision was performed. Language was improved by editing.

 Point 2: Line 15: temperutes, correctly temperatures

 Response 2: The revision was performed. Language was improved by editing.

 Point 3: Line 141: question mark in denominator of equation formula

 Response 3: The revision was performed. Language was improved by editing.

 Point 4: Line 152: unfinished sentence (...measured col...)

 Response 4: The revision was performed. Language was improved by editing.

 Point 5: Line 289: draying, correctly drying

 Response 5: The revision was performed. Language was improved by editing.

Reviewer 2 Report

The manuscriptentitled "Effects of Different Drying Methods on Drying Kinetics, Microstructural, Color, and Rehydration Ratio of the Minced Meat" needs several improvements to be published, especially on the English language. Some corrections are reported in the following comments. The experimental plan is good, even if some results are poorly explained. I suggest minor revisions.

1) English language is very poor and need deep editing from a mother tongue person.

2) Title: remove the article “the” before “Minced Meat”.

3) In my opinion the running Title can be rewritten as “Quality and rehydration properties of differently dried minced meat”.

4) Abstract. Line 12: change the word “firstly” with “for the first time”

5) Abstract. Line 15: change “are” with “were”. In all the text be consistent with the verbal times.

6) Line 20. Please add “s” at the word “ΔE value”, you are talking about more than one value.

7) Line 20. The ΔE ranges that you reported in that way are not clear. You should specify the temperatures of the treatments and compared to which sample they are calculated.

8) Introduction. Line 26. Please change the word “though” with “perceived”.

9) Line 27. Pleas change “food” n “foods” and “of containing” in “they contain”.

10) Line 29. Please change “its” with “their”.

11) Line 32. Please change “due to having” with “being”.

12) Line 32-33. Please modify the sentence as “By drying, the extension of meet and meet products shelf life may be possible thanks to the water activity reduction”.

13) Line 34-35. Please change the sentence as “In addition, the drying process reduce microbial load and cause lighter volume and weight, decreasing storage and transportation cost [5].”

14) Line 39. Please change “meat product” in “meat products”.

15) Line 40. Please remove “properties of the”, it is repeated twice.

16) Line 42. Please modify as “…freeze drying have been used for drying minced meats.”

17) Lines 43-44. Please remove the entire sentence “The conventional drying methods can cause degradation of some properties of food because of heat.” The concept was already expressed before.

18) Lines 44-46. Please change the sentences as “By vacuum drying, the use of low temperature and the absence of oxygen, can preserve heat sensitive and easily oxidizable foods.”

19) Line 49. Please change “should” with “may”.

20) Line 50. Please change “For shortening” in “To shorten”.

21) Please add a reference for the assumption “To shorten the drying time, ultrasonic treatment and vacuum drying are combined in the ultrasound-assisted vacuum drying technique.”

22) Line 51-52. Please change the sentence as “Using this technique, vacuum process, by disrupting food cell walls, accelerates water transfer increasing the drying rate.”

23) Line 55. Please change “in ultrasound drying” with “during ultrasound drying”.

24) Please change “Also” with “Moreover”.

25) Line 61. Please change “its” with “their”

26) Line 61-63. Please change the sentence in “ By freeze drying get products having a porous structure and good rehydration capability, taste, and flavor is possible.”

27) Lines 66-68. Please change the sentence as “In this study vacuum drying (25 °C, 35°C, and 45 °C) without and with ultrasound pretreatment and freeze-drying methods were used to determine the physicochemical properties of minced meat.”

28) Line 72. Please remove “to” after “until”.

29) Had the silicon plate a squared shape? Please specify it.

30) Lines 76-78. Please change the sentence as follows. “The research was carried out in two-steps. Firstly minced meats was dried by different drying techniques, then rehydration ratio, shrinkage, color measurement and microstructure of dried minced meats were determined.”

31) Line 81. Please add the specific number of the AOAC method you cited and add also it in the reference list.

32) Lines 82-84. Please change the sentence as follows. “The drying process was performed by an oven drier (Memmert UF110, Munich, Germany) at 105 °C till constant weight.”

33) Please report “Nielsen (1998)” in the reference list.

34) Lines 85-86. Please change the sentence as follows. “ Hexane was used for the fat extraction.

35) Please remove the dry matter and fat content result from the methods section.

36)Paragraph 2.2.2. In the introduction you said that US was a pretreatment before drying. Is it true? It is not clear from this paragraph. Please explain it better.

37) Please specify the vacuum pressure and the temperature of the freeze drier.

38) Why did you chose 12% as final water content? Please explain it.

39) Paragraph 2.2.3.1. Please specify that MR was calculated only for USV and VD.

40) For each formula you used please write on the side Eq(1), Eq(2), Eq (3)….

41) Line 104. Please change “any time” with “each measurement time”

42) Lines 105-107. Please improve the sentence as follows  “The relative moisture of the drying air (Eq.1), which varies continuously during drying experiments, has been simplified into Eq. (2)”.

43) For equation 3 please explain the meanings of the terms.

44) Lines 117-118. Please change the sentence as follows “For long drying times Eq. (4) can be further simplified in Eq. (5).

45) Line 124. Please add the reference (Doymaz 2013) in the reference list.

46) Lines 130-131. Please change the sentence as follows “2 g of dried minced meat samples were rehydrated in…”

47) Lines 131-132. Please change the sentence as follows “Rehydration ratio is described as the ratio of sample weight after and before rehydration.”

48) Paragraph 2.2.5. How did you measured thickness and diameter of the samples? Please specify it.

49) Line 152. Please change the sentence as follows.” The higher the ΔE value the greater the difference between the two measured samples.”

50) Line 155. You said “before and after rehydration”. You showed only dried samples. Pleas correct.

51) Line 167. Please change the sentence as follows “Figure 1 presented the drying curves (time versus MR ) obtained by ultrasonic vacuum and vacuum drying at 25 ° C, 35 ° C and 45 ° C .”

52) Lines 169. Please change the sentence as follows “….indicating that during this drying period both USV and UV showed constant rate drying.”

53) About the “period” you mentioned earlier, please specify the duration in minutes in which USV and UV showed constant rate drying .

54) Line 172. Please add the word “total” before “drying time”.

55) Lines 172-175. Please change the sentence as follows. “The shortest drying time was observed for the USV drying technique at 45 ° C (82 minutes), this time was approximately  20% shorter  than the one observed for 45° C vacuum drying technique (100 minutes).”

56) Please explain briefly what you observed at  the other temperatures.

57) Lines 175-177. Please change the sentence as follows. “As result of ultrasound, the diffusion boundary layer was changed, and this change can be explained by the properties of the acoustic waves caused by ultrasound[16].

58) Line 178. Please add the word “meat” after chicken and beef.

59) Line 181. Please change the sentence as follows. “As shown in Figure 1, the increase of the drying temperatures decreased significantly the drying times.”

60) Lines 182-183. Please remove completely the sentence “This situation explained that the opposite trend between drying temperatures and drying time. This means that drying rate increases with increasing temperature.”

61) Line 184. Please change the words “with an” with “the”.

62) Did you control the temperature during ultrasound treatment? Do ultrasounds increase the temperature? How did you keep  constant the temperature?

63) Did you measured fats oxidation? Why did you measure total  fat content if you didn’t check oxidation?

64) Figure 1. Please explain the unit for MR.

65) Figure 1 captions. Please correct as. “Drying characteristic of minced meat at different temperatures for vacuum (VD) and ultrasound vacuum drying (USV).” Please add abbreviations for MR. Figures should be self explanatory.

66) Line 199. Remove the word “knowing”.

67) Did you find significant differences between Deff values of VD and USV?

68) Lines 204-205. The sentence is not clear. Please rewrite it.

69) Please organize the images always in the same order first USV and after VD or contrariwise.

70) Figure 3. The figure is not understandable at all. Please reduce the font size of the y axis; please specify what is reported on x axis; please put the legend for all the bars. What does V25, V35 means?

71) Line 211. Please correct. “Table 1 shows…”

72) Line 213. Please change “are” with “resulted”

73) Lines 216-217. Please change the sentence as follows. “The highest rehydration values for  vacuum drying and USV drying technique were obtained at 35° C.”

74) Lines 216-217. You said that the highest rehydration values were obtained at 35° C. Can you explain why? Which is the effect of temperature?

75) Line 217-219. Please modify the sentence as follows “Similarly to our results, in a study [20] on the effect of different drying techniques on some quality parameters of mushrooms, the best rehydration ratio was obtained from freeze-drying.”

76) Lines 221-224. Please modify the sentence as follows. “In a study on  the effect of ultrasound pretreatment on some quality parameters of carrot slices was reported that larger rehydration ratio was observed from the samples subjected to ultrasonic pre-treatment [21].” In this sentence please correct also the font.

77) Lines 227-228. Please correct the sentence as follows. “The higher the rehydration rate, the more water is absorbed resulting closer to the original sample.”

78). Please explain the results in the same order that you used for tables and material and methods. Describe shrinkage before microstructure.

79) Line 230. Please change “under SEM” with “by SEM”.

80) Lines 233-234. Please remove completely the sentence. “Depending on the properties of tissue shrinkage, we can explain the open structure and pore formation.”

81). Line 235. Please remove “and this can be said”.

82) Lines 242-244. You cited Rajewska and Mierzwa [24]. Of which samples are they talking about?

83). Line 246. Please change “…increase with temperature for vacuum and USV drying techniques.”

84) Line 247. Please change “..shrinkage value at higher temperature was also reported in previously published studies..”. Moreover, can you explain the reason of this assumption?

85) Line 248. Please change “ Vacuum technique” with “Vacuum dried meet”

86) Lines 249-250. The sentence “This can explain that ultrasonic treatment effected to porosity structure and for this reason shrinkage values less than vacuum dryer technique.” is not exhaustive other than being not clear.

87) Line 251. Please substitute “by” with “observed after”.

88) Lines 250-251. Can you explain on which samples and in comparison to who Liu et al. reported their results?

89) Add “that” after “than”.

90) Table1. Please add the units. Please report the results in the same order of the text.

91) Table 2. Did you do statistical analysis on these data. Please report it. Who was the reference sample for ΔE calculation? Please report it in the caption.

92) Line 268. Please correct “long time can cause loss of color and darker appearance.”

93) Line 270. Please change “maintained” with “retained”.

94) Line 271-272. Which pigments are responsible for the color change?

95) Line 273. Please remove “of dried minced meat samples”.

96) Line 274 and 277. Please change “maintained” with “retained”.

97) Lines 278-290. In which samples those authors observed that results?

98) Why ultrasound vacuum drying retains color better than vacuum drying? Because of the shorter time? Please explain it.

99) Fig. 5 Caption. Correct “SEM images of the minced meat samples dried at different temperatures and drying methods.”

100)Line 287. Please change “firstly” with “for the first time”.

101) Please add “on minced meat” to the sentence “Ultrasound assisted vacuum drying was for the first time used in this study…”

102) In conclusions you didn’t mentioned freeze drying that according to your results was the best drying technique. The conclusions in my opinion are not complete.

Author Response

Response to Reviewer 2 Comments

 The manuscript entitled "Effects of Different Drying Methods on Drying Kinetics, Microstructural, Color, and Rehydration Ratio of the Minced Meat" needs several improvements to be published, especially on the English language. Some corrections are reported in the following comments. The experimental plan is good, even if some results are poorly explained. I suggest minor revisions.

Thank you very much for your suggestion for improving of our manuscript. The manuscript was extensively revised. All figures were reorganised. The figure caption and abbreviation was revised. The Table 2 (Table 3 in revised version) improved by providing standard deviation and statistic. All abbreviation was checked and corrected (for example Ultrasound assisted vacuum drying is USV, Vacuum drying, VD and freeze drying, FD). Color figures reorganised and quality of the figures was improved. Peroxide analysis results were presented and discussed. Introduction part and discussion part were improved. Manuscript was checked for language, revised based on the reviewer recommendations and sent to language editing services.  

 Point 1:  English language is very poor and need deep editing from a mother tongue person.

Response: Language was revised based on the reviewer and editing services suggestions. 

Point  2) Title: remove the article “the” before “Minced Meat”.

Response: The revision was performed according to reviewer suggestions.

Point 3) In my opinion the running Title can be rewritten as “Quality and rehydration properties of differently dried minced meat”.

Response: The revision was performed according to reviewer suggestions.

Point 4) Abstract. Line 12: change the word “firstly” with “for the first time”

Response: The revision was performed according to reviewer suggestions.

Point 5) Abstract. Line 15: change “are” with “were”. In all the text be consistent with the verbal times.

Response: The revision was performed according to reviewer suggestions.

Point 6) Line 20. Please add “s” at the word “ΔE value”, you are talking about more than one value.

Response: The revision was performed according to reviewer suggestions.

 Point 7) Line 20. The ΔE ranges that you reported in that way are not clear. You should specify the temperatures of the treatments and compared to which sample they are calculated.

Response: The revision was performed according to reviewer suggestions. The delta E values were presented for all temperature.

Point 8) Introduction. Line 26. Please change the word “though” with “perceived”.

Response: The revision was performed according to reviewer suggestions.

Point 9) Line 27. Pleas change “food” n “foods” and “of containing” in “they contain”.

Response: The revision was performed according to reviewer suggestions.

Point 10) Line 29. Please change “its” with “their”.

Response: The revision was performed according to reviewer suggestions.

Point 11) Line 32. Please change “due to having” with “being”.

Response: The revision was performed according to reviewer suggestions.

Point 12) Line 32-33. Please modify the sentence as “By drying, the extension of meet and meet products shelf life may be possible thanks to the water activity reduction”.

Response: The revision was performed according to reviewer suggestions.

Point 13) Line 34-35. Please change the sentence as “In addition, the drying process reduce microbial load and cause lighter volume and weight, decreasing storage and transportation cost [5].”

Response: The revision was performed according to reviewer suggestions.

Point 14) Line 39. Please change “meat product” in “meat products”.

Response: The revision was performed according to reviewer suggestions.

Point 15) Line 40. Please remove “properties of the”, it is repeated twice.

Response: The revision was performed according to reviewer suggestions.

Point 16) Line 42. Please modify as “…freeze drying have been used for drying minced meats.”

Response: The revision was performed according to reviewer suggestions.

Point 17) Lines 43-44. Please remove the entire sentence “The conventional drying methods can cause degradation of some properties of food because of heat.” The concept was already expressed before.

Response: The revision was performed according to reviewer suggestions.

Point 18) Lines 44-46. Please change the sentences as “By vacuum drying, the use of low temperature and the absence of oxygen, can preserve heat sensitive and easily oxidizable foods.”

Response: The revision was performed according to reviewer suggestions.

Point 19) Line 49. Please change “should” with “may”.

Response: The revision was performed according to reviewer suggestions.

Point 20) Line 50. Please change “For shortening” in “To shorten”.

Response: The revision was performed according to reviewer suggestions.

Point 21) Please add a reference for the assumption “To shorten the drying time, ultrasonic treatment and vacuum drying are combined in the ultrasound-assisted vacuum drying technique.”

Point 22) Line 51-52. Please change the sentence as “Using this technique, vacuum process, by disrupting food cell walls, accelerates water transfer increasing the drying rate.”

Response: The revision was performed according to reviewer suggestions.

Point 23) Line 55. Please change “in ultrasound drying” with “during ultrasound drying”.

Response: The revision was performed according to reviewer suggestions.

Point 24) Please change “Also” with “Moreover”.

Response: The revision was performed according to reviewer suggestions.

Point 25) Line 61. Please change “its” with “their”

Response: The revision was performed according to reviewer suggestions.

Point 26) Line 61-63. Please change the sentence in “ By freeze drying get products having a porous structure and good rehydration capability, taste, and flavor is possible.”

Response: The revision was performed according to reviewer suggestions.

Point 27) Lines 66-68. Please change the sentence as “In this study vacuum drying (25 °C, 35°C, and 45 °C) without and with ultrasound pretreatment and freeze-drying methods were used to determine the physicochemical properties of minced meat.”

Response: The revision was performed according to reviewer suggestions.

Point 28) Line 72. Please remove “to” after “until”.

Response: The revision was performed according to reviewer suggestions.

Point 29) Had the silicon plate a squared shape? Please specify it.

Point 30) Lines 76-78. Please change the sentence as follows. “The research was carried out in two-steps. Firstly minced meats was dried by different drying techniques, then rehydration ratio, shrinkage, color measurement and microstructure of dried minced meats were determined.”

Response: The revision was performed according to reviewer suggestions.

Point 31) Line 81. Please add the specific number of the AOAC method you cited and add also it in the reference list.

Response: specific number of the AOAC for dry matter analysis were provided. Official methods were also provided for proxide and total fat content analysis. 

Point 32) Lines 82-84. Please change the sentence as follows. “The drying process was performed by an oven drier (Memmert UF110, Munich, Germany) at 105 °C till constant weight.”

Response: The revision was performed according to reviewer suggestions.

Point 33) Please report “Nielsen (1998)” in the reference list.

Response: The references was removed. Official methods was provided.

Point 34) Lines 85-86. Please change the sentence as follows. “ Hexane was used for the fat extraction.

Response: The revision was performed according to reviewer suggestions.

Point 35) Please remove the dry matter and fat content result from the methods section.

Response: The revision was performed according to reviewer suggestions.

Point 36)Paragraph 2.2.2. In the introduction you said that US was a pretreatment before drying. Is it true? It is not clear from this paragraph. Please explain it better.

Response: The sentence was revised. The methods for USV drying was provided.

 Point 37) Please specify the vacuum pressure and the temperature of the freeze drier.

Response: Temperature and vacuum pressure values were provided.

Point 38) Why did you chose 12% as final water content? Please explain it.

Response: Provided in material method section.

Point 39) Paragraph 2.2.3.1. Please specify that MR was calculated only for USV and VD.

Response: The revision was performed according to reviewer suggestions.

Point 40) For each formula you used please write on the side Eq(1), Eq(2), Eq (3)….

Response: The revision was performed according to reviewer suggestions.

Point 41) Line 104. Please change “any time” with “each measurement time”

Response: The revision was performed according to reviewer suggestions.

Point 42) Lines 105-107. Please improve the sentence as follows  “The relative moisture of the drying air (Eq.1), which varies continuously during drying experiments, has been simplified into Eq. (2)”.

Response: The revision was performed according to reviewer suggestions.

Point 43) For equation 3 please explain the meanings of the terms.

Response: The revision was performed according to reviewer suggestions.

Point 44) Lines 117-118. Please change the sentence as follows “For long drying times Eq. (4) can be further simplified in Eq. (5).

Response: The revision was performed according to reviewer suggestions.

Point 45) Line 124. Please add the reference (Doymaz 2013) in the reference list.

Response: The references was provided.

Point 46) Lines 130-131. Please change the sentence as follows “2 g of dried minced meat samples were rehydrated in…”

Response: The revision was performed according to reviewer suggestions.

Point 47) Lines 131-132. Please change the sentence as follows “Rehydration ratio is described as the ratio of sample weight after and before rehydration.”

Response: The revision was performed according to reviewer suggestions.

Point 48) Paragraph 2.2.5. How did you measured thickness and diameter of the samples? Please specify it.

Response: The revision was performed according to reviewer suggestions. The measurement methods of diameter and thickness was provided.

Point 49) Line 152. Please change the sentence as follows.” The higher the ΔE value the greater the difference between the two measured samples.”

Response: The revision was performed according to reviewer suggestions.

Point 50) Line 155. You said “before and after rehydration”. You showed only dried samples. Pleas correct.

Response: The revision was performed according to reviewer suggestions.

Point 51) Line 167. Please change the sentence as follows “Figure 1 presented the drying curves (time versus MR ) obtained by ultrasonic vacuum and vacuum drying at 25 ° C, 35 ° C and 45 ° C .”

Response: The revision was performed according to reviewer suggestions.

Point 52) Lines 169. Please change the sentence as follows “….indicating that during this drying period both USV and UV showed constant rate drying.”

Response: The revision was performed according to reviewer suggestions.

Point 53) About the “period” you mentioned earlier, please specify the duration in minutes in which USV and UV showed constant rate drying .

Response: The duration times in constant period at all drying temperatures were provided.

Point 54) Line 172. Please add the word “total” before “drying time”.

Response: The revision was performed according to reviewer suggestions.

Point 55) Lines 172-175. Please change the sentence as follows. “The shortest drying time was observed for the USV drying technique at 45 ° C (82 minutes), this time was approximately  20% shorter  than the one observed for 45° C vacuum drying technique (100 minutes).”

Response: The revision was performed according to reviewer suggestions.

Point 56) Please explain briefly what you observed at  the other temperatures.

Response: Provided for drying temperature.

Point 57) Lines 175-177. Please change the sentence as follows. “As result of ultrasound, the diffusion boundary layer was changed, and this change can be explained by the properties of the acoustic waves caused by ultrasound[16].

Response: The revision was performed according to reviewer suggestions.

Point 58) Line 178. Please add the word “meat” after chicken and beef.

Response: The revision was performed according to reviewer suggestions.

Point 59) Line 181. Please change the sentence as follows. “As shown in Figure 1, the increase of the drying temperatures decreased significantly the drying times.”

Response: The revision was performed according to reviewer suggestions.

Point 60) Lines 182-183. Please remove completely the sentence “This situation explained that the opposite trend between drying temperatures and drying time. This means that drying rate increases with increasing temperature.”

Response: The revision was performed according to reviewer suggestions.

Point 61) Line 184. Please change the words “with an” with “the”.

Response: The revision was performed according to reviewer suggestions.

Point 62) Did you control the temperature during ultrasound treatment? Do ultrasounds increase the temperature? How did you keep  constant the temperature?

The USV system is a combination of an ultrasound water bath and a vacuum pump. This system was

detailed in our previous published studies (Başlar, Kilicli, et al., 2014; Başlar, Kiliçli, & Yalinkiliç, 2015). The samples were put into a conical flask attached to the vacuum pump and sonicated using the ultrasonic water bath. In this system, after the temperature is regulated by ultrasonic water bath, drying starts and no significant deviation in temperature is observed during drying.

Baslar, M.; Kilicli, M.; Yalinkilic, B. Dehydration kinetics of salmon and trout fillets using ultrasonic vacuum drying as a novel technique. Ultrasonics Sonochemistry 2015, 27, 495-502, doi:10.1016/j.ultsonch.2015.06.018.

Point 63) Did you measured fats oxidation? Why did you measure total  fat content if you didn’t check oxidation?

Response: Peroxide analysis was conducted. Results were presented in Table 1 and discussed.

Point 64) Figure 1. Please explain the unit for MR.

Response: The revision was performed according to reviewer suggestions.

Point 65) Figure 1 captions. Please correct as. “Drying characteristic of minced meat at different temperatures for vacuum (VD) and ultrasound vacuum drying (USV).” Please add abbreviations for MR. Figures should be self explanatory.

Response: The revision was performed according to reviewer suggestions.

Point 66) Line 199. Remove the word “knowing”.

Response: The revision was performed according to reviewer suggestions.

Point 67) Did you find significant differences between Deff values of VD and USV?

Response: for 35 and 45 C results were significiant but for 25 Deff results is insignificant.

Point 68) Lines 204-205. The sentence is not clear. Please rewrite it.

Response: The revision was performed according to reviewer suggestions.

Point 69) Please organize the images always in the same order first USV and after VD or contrariwise.

Reponse: All figures were reorganized. Fig caption revised and visual quality was improved. 

Point 70) Figure 3. The figure is not understandable at all. Please reduce the font size of the y axis; please specify what is reported on x axis; please put the legend for all the bars. What does V25, V35 means?

Response: All figures were reorganized. For this Fig, information about axis and fig captions were revised.

Point 71) Line 211. Please correct. “Table 1 shows…”

Revision was performed.

Point 72) Line 213. Please change “are” with “resulted”

Revision was performed.

Point 73) Lines 216-217. Please change the sentence as follows. “The highest rehydration values for  vacuum drying and USV drying technique were obtained at 35° C.”

Reponse:  Revision was performed according to reviewer suggestions. 

Point 74) Lines 216-217. You said that the highest rehydration values were obtained at 35° C. Can you explain why? Which is the effect of temperature?

Response: Temperature and time combination (moderate and time temperature) could cause high rehydration.

Point 75) Line 217-219. Please modify the sentence as follows “Similarly to our results, in a study [20] on the effect of different drying techniques on some quality parameters of mushrooms, the best rehydration ratio was obtained from freeze-drying.”

Reponse:  Revision was performed according to reviewer suggestions. 

Point 76) Lines 221-224. Please modify the sentence as follows. “In a study on  the effect of ultrasound pretreatment on some quality parameters of carrot slices was reported that larger rehydration ratio was observed from the samples subjected to ultrasonic pre-treatment [21].” In this sentence please correct also the font.

Reponse:  Revision was performed according to reviewer suggestions. 

Point 77) Lines 227-228. Please correct the sentence as follows. “The higher the rehydration rate, the more water is absorbed resulting closer to the original sample.”

Reponse:  Revision was performed according to reviewer suggestions. 

Point 78). Please explain the results in the same order that you used for tables and material and methods. Describe shrinkage before microstructure.

Reponse:  Revision was performed according to reviewer suggestions.

Point 79) Line 230. Please change “under SEM” with “by SEM”.

Reponse:  Revision was performed according to reviewer suggestions. 

Point 80) Lines 233-234. Please remove completely the sentence. “Depending on the properties of tissue shrinkage, we can explain the open structure and pore formation.”

Reponse:  Revision was performed according to reviewer suggestions. 

Point 81). Line 235. Please remove “and this can be said”.

Reponse:  Revision was performed according to reviewer suggestions. 

Point 82) Lines 242-244. You cited Rajewska and Mierzwa [24]. Of which samples are they talking about?

Response: The samples for drying was provided.

Point 83). Line 246. Please change “…increase with temperature for vacuum and USV drying techniques.”

Reponse:  Revision was performed according to reviewer suggestions. 

Point 84) Line 247. Please change “..shrinkage value at higher temperature was also reported in previously published studies..”. Moreover, can you explain the reason of this assumption?

Response. New references was provided and discussed.

85) Line 248. Please change “ Vacuum technique” with “Vacuum dried meat”

Reponse:  Revision was performed according to reviewer suggestions. 

Point 86) Lines 249-250. The sentence “This can explain that ultrasonic treatment effected to porosity structure and for this reason shrinkage values less than vacuum dryer technique.” is not exhaustive other than being not clear.

Response: This sentences was removed. Discussed by new references.

Point 87) Line 251. Please substitute “by” with “observed after”.

Reponse:  Revision was performed according to reviewer suggestions. 

Point 88) Lines 250-251. Can you explain on which samples and in comparison to who Liu et al. reported their results?

Response: Samples was provided.

Point 89) Add “that” after “than”.

Reponse:  Revision was performed according to reviewer suggestions. 

Point 90) Table1. Please add the units. Please report the results in the same order of the text.

Reponse:  Revision was performed according to reviewer suggestions. 

Point 91) Table 2. Did you do statistical analysis on these data. Please report it. Who was the reference sample for ΔE calculation? Please report it in the caption.

Response: statistical analysis and Sd value were presented. The table related color was improved.

Point 92) Line 268. Please correct “long time can cause loss of color and darker appearance.”

Reponse:  Revision was performed according to reviewer suggestions. 

Point 93) Line 270. Please change “maintained” with “retained”.

Reponse:  Revision was performed according to reviewer suggestions. 

Point 94) Line 271-272. Which pigments are responsible for the color change?

Response: This sentence was revised.

Point 95) Line 273. Please remove “of dried minced meat samples”.

Reponse:  Revision was performed according to reviewer suggestions. 

Point 96) Line 274 and 277. Please change “maintained” with “retained”.

Reponse:  Revision was performed according to reviewer suggestions. 

Point 97) Lines 278-290. In which samples those authors observed that results?

Response: sample was provided.

Point 98) Why ultrasound vacuum drying retains color better than vacuum drying? Because of the shorter time? Please explain it.

Response: discussion for color change was improved.

Point 99) Fig. 5 Caption. Correct “SEM images of the minced meat samples dried at different temperatures and drying methods.”

Reponse:  Revision was performed according to reviewer suggestions. 

Point 100)Line 287. Please change “firstly” with “for the first time”.

Reponse:  Revision was performed according to reviewer suggestions. 

Point 101) Please add “on minced meat” to the sentence “Ultrasound assisted vacuum drying was for the first time used in this study…”

Reponse:  Revision was performed according to reviewer suggestions. 

Point 102) In conclusions you didn’t mentioned freeze drying that according to your results was the best drying technique. The conclusions in my opinion are not complete.

Response: Conclusion part was improved by providing FD and PV results.

Reviewer 3 Report

The research topic presented for review of the manuscript is undoubtedly very interesting. However, I have several critical remarks about its content. It is worth noting that the whole of the manuscript is prepared carelessly in terms of both editorial and content-related aspects.

In the entire content of the manuscript there are many spaces used in an unjustified way, you can find repetitions of words whose spelling are not correct. The all manuscript is bad presented taking into account style and grammar.

The INTRODUCTION  (Chapter 1) does not explain why the authors undertook the discussed research topic. The presented introduction do not explain why there is a need to optimize the methods and parameters of meat drying. There is also no explanation in the text where you can use the received dried products and whether drying in the case of minced meat is just a method of preservation or a way to obtain an innovative semi-finished product that can be used as a component of ready meals.

The methodology lacks references to literature.

The name of Chapter 3 RESULTS AND DISCUSSION suggests that the authors should not only present the obtained results but also do the discussion with the research presented in the latest literature - unfortunately, in the text of the reviewed manuscript, there is no reliable discussion of the results. Lack of a discussion of the presented results limits the possibility of presenting good and correct conclusions.

Numerous formal errors (some distinguished in the text) and the lack of discussion make me sadly reject the submitted manuscript.

Author Response

Response to Reviewer 3 Comments

 Comments: The research topic presented for review of the manuscript is undoubtedly very interesting. However, I have several critical remarks about its content. It is worth noting that the whole of the manuscript is prepared carelessly in terms of both editorial and content-related aspects. In the entire content of the manuscript there are many spaces used in an unjustified way, you can find repetitions of words whose spelling are not correct. The all manuscript is bad presented taking into account style and grammar. The INTRODUCTION (Chapter 1) does not explain why the authors undertook the discussed research topic. The presented introduction do not explain why there is a need to optimize the methods and parameters of meat drying. There is also no explanation in the text where you can use the received dried products and whether drying in the case of minced meat is just a method of preservation or a way to obtain an innovative semi-finished product that can be used as a component of ready meals.The methodology lacks references to literature.The name of Chapter 3 RESULTS AND DISCUSSION suggests that the authors should not only present the obtained results but also do the discussion with the research presented in the latest literature - unfortunately, in the text of the reviewed manuscript, there is no reliable discussion of the results. Lack of a discussion of the presented results limits the possibility of presenting good and correct conclusions. Numerous formal errors (some distinguished in the text) and the lack of discussion make me sadly reject the submitted manuscript.

Response: Thank you very much for your suggestion for improving of our manuscript. The manuscript was extensively revised. All figures were reorganised. The figure caption and abbreviation was revised. The Table 2 (Table 3 in revised version) improved by providing standart deviation and statistic. All abbreviation was checked and corrected (for example Ultrasound assisted vacuum dryin is USV, Vacumm drying VD and freeze drying FD). Color figures reorganised and quality of the figures was improved. Peroxide analysis results were presented and discussed. Introduction part and discussion part were improved. Manuscript was sent to language editing.

Round  2

Reviewer 3 Report

The authors of the manuscript presented for the evaluation made numerous changes in the text. The attempt to improve the manuscript has improved its substantive value.

Author Response

Comments:

Comments and Suggestions for Authors: The authors of the manuscript presented for the evaluation made numerous changes in the text. The attempt to improve the manuscript has improved its substantive value.

Response: Thank you very much for your contrubution on our manuscript revison. The manuscript was extensvively revised in First revison. Introduction part was improved for the explaining of drying methods and parameters on meat products drying. Material methods section was extended by detailing methods parts. Figüres and tables were also be improved and new references were used for discussing. During second revison, material methods and discussion part were also improved by new citing new references. for example,

-2.2.2, new references was cited for vacuum drying methods,

-In feeze drying part, method was detailed by providing drying temperature, vacuum value and drying time.

-2.2.3.1. New methods was inserted for moisture ratio,

-2.2.3.2New references were provided for Deff value [20,21],activation energy calculation [23]

-2.2.3.4. Sopporting references were provided  [30, 31]
